# Evidence for evolutionary divergence of activity-dependent gene expression in developing neurons

Jing Qiu[1], Jamie McQueen[1], Bilada Bilican[2], Owen Dando[1,3], Dario Magnani[2], Karolina Punovuori[2], Bhuvaneish T Selvaraj[2], Matthew Livesey[1], Ghazal Haghi[1,2], Samuel Heron[4], Karen Burr[2], Rickie Patani[2], Rinku Rajan[1,2], Olivia Sheppard[5], Peter C Kind[1,3], T Ian Simpson[4], Victor LJ Tybulewicz[6,7], David JA Wyllie[1], Elizabeth MC Fisher[5], Sally Lowell[2], Siddharthan Chandran[2,3]*, Giles E Hardingham[1]*

[1]School of Biomedical Sciences, University of Edinburgh, Edinburgh, United Kingdom; [2]MRC Centre for Regenerative Medicine, University of Edinburgh, Edinburgh, United Kingdom; [3]Centre for Brain Development and Repair, Institute for Stem Cell Biology and Regenerative Medicine, National Centre for Biological Sciences, Bangalore, India; [4]School of Informatics, University of Edinburgh, Edinburgh, United Kingdom; [5]Department of Neurodegenerative Disease, UCL Institute of Neurology, London, United Kingdom; [6]The Francis Crick Institute, London, United Kingdom; [7]Imperial College, London, United Kingdom

*For correspondence:
siddharthan.chandran@ed.ac.uk
(SC); Giles.Hardingham@ed.ac.uk
(GEH)

Competing interests: The authors declare that no competing interests exist.

**Abstract** Evolutionary differences in gene regulation between humans and lower mammalian experimental systems are incompletely understood, a potential translational obstacle that is challenging to surmount in neurons, where primary tissue availability is poor. Rodent-based studies show that activity-dependent transcriptional programs mediate myriad functions in neuronal development, but the extent of their conservation in human neurons is unknown. We compared activity-dependent transcriptional responses in developing human stem cell-derived cortical neurons with those induced in developing primary- or stem cell-derived mouse cortical neurons. While activity-dependent gene-responsiveness showed little dependence on developmental stage or origin (primary tissue vs. stem cell), notable species-dependent differences were observed. Moreover, differential species-specific gene ortholog regulation was recapitulated in aneuploid mouse neurons carrying human chromosome-21, implicating promoter/enhancer sequence divergence as a factor, including human-specific activity-responsive AP-1 sites. These findings support the use of human neuronal systems for probing transcriptional responses to physiological stimuli or indeed pharmaceutical agents.

## Introduction

The last common ancestor of mice and humans existed around 80 million years ago, sufficient time for divergence in signal-dependent gene regulation (*Villar et al., 2014*). However, the conservation or divergence of dynamic signal-dependent programs of gene expression is incompletely understood, particularly in the nervous system. A fundamental transcriptional program of this sort is that elicited in neurons by electrical activity, triggered via $Ca^{2+}$ influx through ligand- and/or voltage-gated $Ca^{2+}$ channels and activating genes containing $Ca^{2+}$-responsive transcription factor binding sites in their promoters (*Sheng and Greenberg, 1990*; *Morgan and Curran, 1991*; *West et al., 2001*). These changes in gene expression are critical to a neuron's functional response to electrical

**eLife digest** Cells in the brain known as neurons produce electrical activity that allows them to signal to other cells; this in turn allows us to think, feel, move, remember and learn. This electrical activity also causes the neurons to increase or decrease the activity of certain genes. Whether a gene is controlled by electrical activity depends on the structure of the regions of DNA, called promoters or enhancers, where certain proteins can bind in order to activate the gene.

Studying how human neurons respond to electrical stimulation has been challenging because they are difficult to obtain and grow in the laboratory. Instead, most experiments have been conducted with mouse or rat neurons. However, it was not known to what extent the changes seen in gene activity in rodent neurons reflect the changes that occur in their human equivalents.

Qiu et al. grew human neurons from human embryonic stem cells and compared these cells with the corresponding mouse neurons. Overall, the cells from both species generally reacted similarly to simulated electrical stimulation. However, some genes responded in significantly different ways in mouse and human neurons. For example, a gene called *ETS2* was switched on more quickly and strongly in the human neurons than in mouse neurons. Further experiments indicated that some of these differences are because the promoter and enhancer regions of the genes have evolved in different ways in mice and humans.

More research is now needed to test whether the differences in gene activation seen in the mouse and human neurons in response to electrical activity affect how the neurons work. It will be equally important to investigate whether neurons from different species respond differently to other factors, such as drugs.

activity in development and maturity (*Konur and Ghosh, 2005*; *West and Greenberg, 2011*; *Bell and Hardingham, 2011*), and are totally distinct from the toxic sequelae of excitotoxic $Ca^{2+}$ influx (*Hardingham and Bading, 2010*; *Wyllie et al., 2013*). For example, rodent studies have shown that activity-dependent gene expression programs direct myriad processes in developing neurons, including neuroprotection, dendritic arborization, and synaptic plasticity (*Konur and Ghosh, 2005*; *West and Greenberg, 2011*; *Bell and Hardingham, 2011*). Moreover, certain neuro-developmental disorders are associated with defects in activity-dependent transcriptional mechanisms (*West and Greenberg, 2011*), making it important to fully understand transcriptional programs that are triggered by electrical activity in developing human neurons.

Comparing the influence of neuronal electrical activity on mouse-human orthologs, and identifying the basis for any differences, requires a combination of approaches, given the inability to reproducibly study primary human developing neurons. Embryonic stem cell (ESC)-based technology enables the generation of glutamatergic cortical-patterned neurons from human embryonic stem cells (hESC$^{CORT}$-neurons) of sufficient homogeneity and electrical maturity (*Bilican et al., 2014*; *Livesey et al., 2014*) to enable the study of activity-dependent gene expression. Such responses can then be compared to both primary mouse cortical neurons (of differing developmental stages) as well as those derived from mouse ESCs, to distinguish species-specific differences from those dependent on developmental stage or origin (primary tissue vs. stem cell line). Moreover, it is in theory possible to study mouse-human ortholog regulation in the same neuron by exploiting the aneuploid Tc1 mouse which carries a freely segregating copy of human chromosome-21, to identify whether any differences are independent of cellular environment (i.e. are due to DNA sequence divergence). A combination of these approaches has been employed to reveal strong conservation of the neuronal activity-dependent transcriptome onto which a significant degree of divergence is overlaid.

## Results and discussion

We generated dissociated glutamatergic cortical-patterned neurons from human embryonic stem cells (hESC$^{CORT}$-neurons), whose characterization is described fully elsewhere (*Bilican et al., 2014*; *Livesey et al., 2014*). These cells have a combination of homogeneity and electrical maturity that is hard to achieve with classical human stem cell-derived neurosphere-based preparations, which have

a poor dynamic range in terms of gene regulation (*Paşca et al., 2011*) compared to primary mouse neuronal preparations (*Spiegel et al., 2014*)

We studied transcriptional responses to L-type $Ca^{2+}$ channel activation, an important mediator of activity-dependent gene regulation (*West and Greenberg, 2011*; *Sheng et al., 1990*; *Bito et al., 1997*; *Deisseroth et al., 2003*; *Wheeler et al., 2012*; *Ma et al., 2014*). To do this, hESC$^{CORT}$-neurons were subject to KCl-induced membrane depolarization in the presence of the L-type $Ca^{2+}$ channel agonist FPL64176 (KCl/FPL), which leads to robust and uniform $Ca^{2+}$ responses ([*Bilican et al., 2014*], *Figure 1—figure supplement 1a*, *Figure 1—source data 1*) similar to rodent neurons (*Hardingham et al., 1999*). QPCR analysis revealed robust induction of classical early-response activity-regulated genes (ARGs) *FOS, FOSL2 (FRA-2)*, and *FOSB*, and neurobiologically important ARGs *BDNF, ARC*, and *NPAS4* (*Figure 1—figure supplement 1b*, *Figure 1—source data 3*) whose regulation has hitherto been studied primarily in rodent neurons.

We then performed RNA-seq analysis of KCl/FPL-induced gene expression changes in hESC$^{CORT}$-neurons as well as days-in-vitro (DIV)10 mouse primary cortical neurons (Mus-PRIM$^{CORT}$-neurons, predominantly excitatory, like Hum-ESC$^{CORT}$-neurons), focusing on a 4h timepoint (*Figure 1a,b*, *Figure 1—source data 1*). QPCR validation of fold-induction of selected ARGs revealed a tight correlation with the RNA-seq data (r > 0.99 for Hum-ESC$^{CORT}$-neurons and mouse neurons respectively, *Figure 1—figure supplement 1c, 1d*, *Figure 1—source data 4*), supporting the reliability of the RNA-seq data set. Our interspecies comparisons were restricted to 1:1 orthologous pairs whose average expression level met a minimum threshold (11,302 genes expressed >0.5 FPKM on average, in all cell types). Genes induced >5-fold in both Hum-ESC$^{CORT}$- and DIV10 Mus-PRIM$^{CORT}$-neurons according to RNA-seq included neurobiologically important genes such as *PER1, EGR2, EGR4* and *ATF3* (*Figure 1c*). Global comparison of $Log_2$ (gene fold-change) in orthologous pairs revealed a correlation between Hum-ESC$^{CORT}$-neurons and DIV10 Mus-PRIM$^{CORT}$-neurons (r = 0.480, 95% CI: 0.465 to 0.494, p<0.0001, *Figure 1d*, *Figure 1—source data 1*), pointing to a significant, but incomplete, degree of conservation. For the 11,302 ortholog pairs, we calculated the 'differential regulation index' (DRI), defined as the fold-change (Hum-ESC$^{CORT}$-neurons) divided by the fold-change (Mus-PRIM$^{CORT}$-neurons). The $Log_2$ of the DRIs are normally distributed about zero with a standard deviation of 0.68 (*Figure 1—figure supplement 1e*, *Figure 1—source data 5*).

While this imperfect conservation could involve species-specific gene responsiveness, it could be due to other factors: (i) non-developmental equivalency in the human and mouse neuronal preparations combined with a dependency of the responsiveness of certain genes on developmental stage could underlie differences; (ii) if gene-responsiveness were sensitive to the origin of the neurons studied (embryonic stem cell line (human) vs. primary tissue (mouse)) this could also lead to differences, and an erroneous exaggeration of apparent evolutionary divergence. To address (i) we compared KCl/FPL-induced gene regulation in DIV10 Mus-PRIM$^{CORT}$-neurons with that in synaptically immature DIV4 Mus-PRIM$^{CORT}$-neurons, which revealed a strong correlation (p<0.0001, r = 0.834, 95% CI: 0.833 to 0.844, *Figure 1e*, *Figure 1—source data 1*), and a $log_2$ DRI (DIV4:DIV10) distribution curve (*Figure 1—figure supplement 1f*, *Figure 1—source data 5*) substantially narrower than that observed in *Figure 1—figure supplement 1e*. Taken together, this suggests that developmental stage does not have a large effect on the activity-responsiveness of genes at this time point and is unlikely to account for the human-mouse differences observed in *Figure 1d*.

To address (ii) (i.e. whether stem cell- vs. primary tissue origin causes differential gene responsiveness) we generated cortical-patterned neurons from mouse embryonic stem cells (Mus-ESC$^{CORT}$-neurons) using established protocols (*Gaspard et al., 2008*), which exhibit spontaneous firing and synaptic activity (*Figure 2a–d*), similar to our Hum-ESC$^{CORT}$-neurons (*Bilican et al., 2014*; *Livesey et al., 2014*). The Mus-ESC$^{CORT}$-neurons were subject to the same 4h KCl/FPL stimulation and RNA-seq analysis (*Figure 2—figure supplement 1a*). Same-species comparison of gene fold-change was made between Mus-PRIM$^{CORT}$-neurons (DIV4 and DIV10) and Mus-ESC$^{CORT}$-neurons in the 11,302 genes compared in *Figure 1e,f*. A strong correlation was observed in gene-responsiveness in Mus-ESC$^{CORT}$-neurons vs. Mus-PRIM$^{CORT}$-neurons at either DIV10 or DIV4 (p<0.0001, r = 0.748 (DIV10), *Figure 2e*; r = 0.788, (DIV4), *Figure 2f*), significantly stronger than between Mus-ESC$^{CORT}$- and Hum-ESC$^{CORT}$-neurons (r = 0.596, *Figure 2g*, see also *Figure 2—source data 1*). Indeed, a comparison of gene fold-change between all biological replicates in all data sets revealed that all mouse neuronal preparations (primary DIV10, primary DIV4, and ESC-derived) correlated far more strongly with each other than with the human ESC$^{CORT}$-neurons. (*Figure 2h*). This suggests

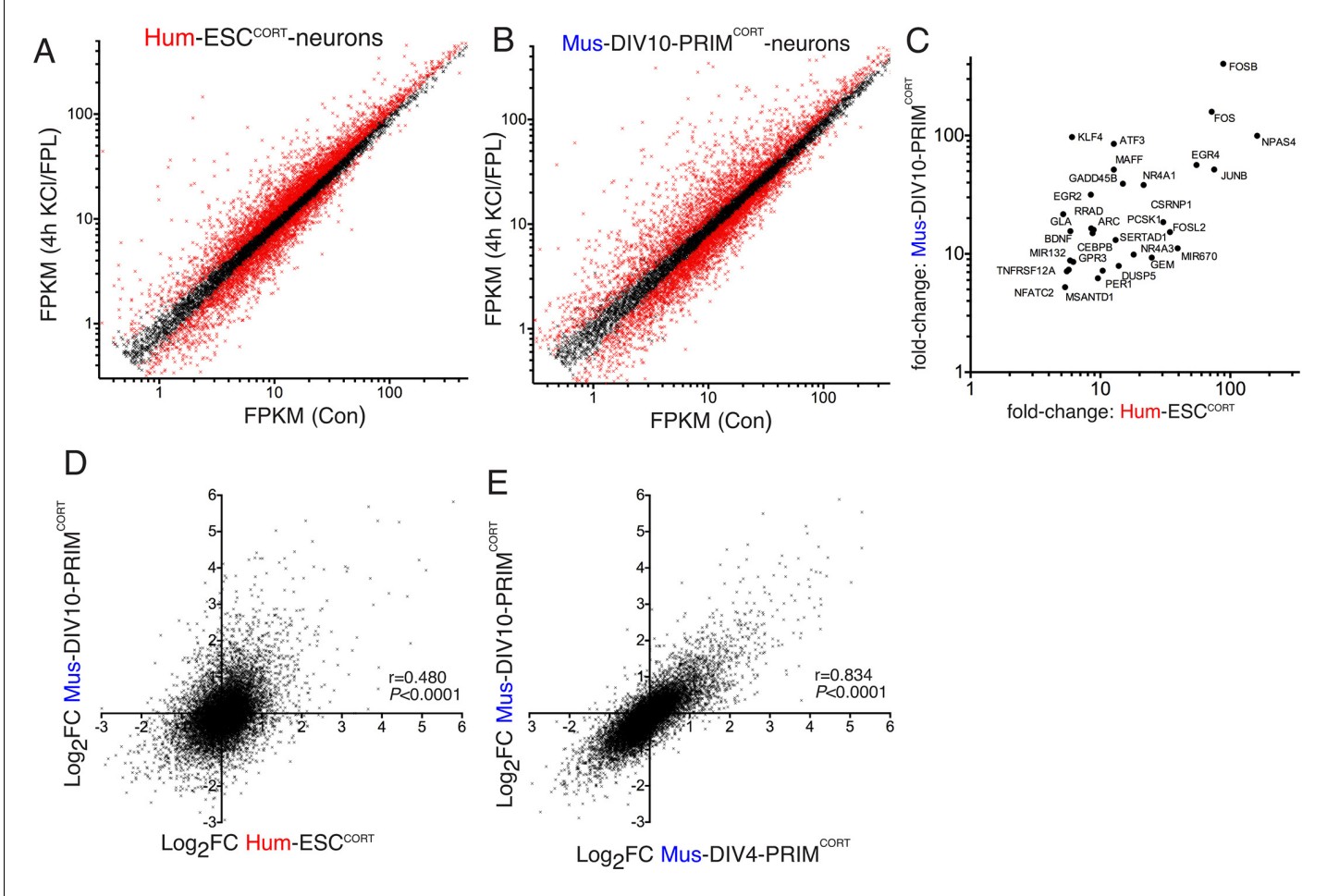

**Figure 1.** Conservation and divergence in gene regulation in neurons of human and mouse origin. (**A**,**B**) Analysis of gene expression changes induced by KCl/FPL in Hum-ESC$^{CORT}$-neurons (**A**) and DIV10 Mus-PRIM$^{CORT}$-neurons (**B**). Normalised RNA-seq read density (FPKM) mapping to each gene in RNA extracted from control vs. KCl/FPL-treated neurons is shown (n = 3 independent biological replicates). Genes whose expression was significantly altered by KCl/FPL treatment (Benjamini-Hochberg-adjusted p-value<0.05, calculated within DESeq2) are highlighted in red. (**C**) Cohort of human: mouse orthologous pairs where both are strongly (>5-fold) and significantly (Benjamini-Hochberg-adjusted p-value<0.05) induced in Hum-ESC$^{CORT}$-neurons and DIV10 Mus-PRIM$^{CORT}$-neurons respectively. (**D**) Correlation of KCl/FPL-induced fold-change in 11,302 ortholog pairs in DIV10 Mus-PRIM$^{CORT}$-neurons vs. DIV10-Hum-ESC$^{CORT}$-neurons. (**E**) Correlation of KCl/FPL-induced fold-change in the same 11,302 genes as in (**D**) in DIV10 vs. DIV4 Mus-PRIM$^{CORT}$-neurons.

The following source data and figure supplement are available for figure 1:

**Source data 1.** Data set relating to *Figure 1a–e*.
**Source data 2.** Data set relating to *Figure 1—figure supplement 1a*.
**Source data 3.** Data set relating to *Figure 1—figure supplement 1b*.
**Source data 4.** Data set relating to *Figure 1—figure supplement 1c–d*.
**Source data 5.** Data set relating to *Figure 1—figure supplement 1e–f*.
**Figure supplement 1.** Conservation and divergence in gene regulation in neurons of human and mouse origin.

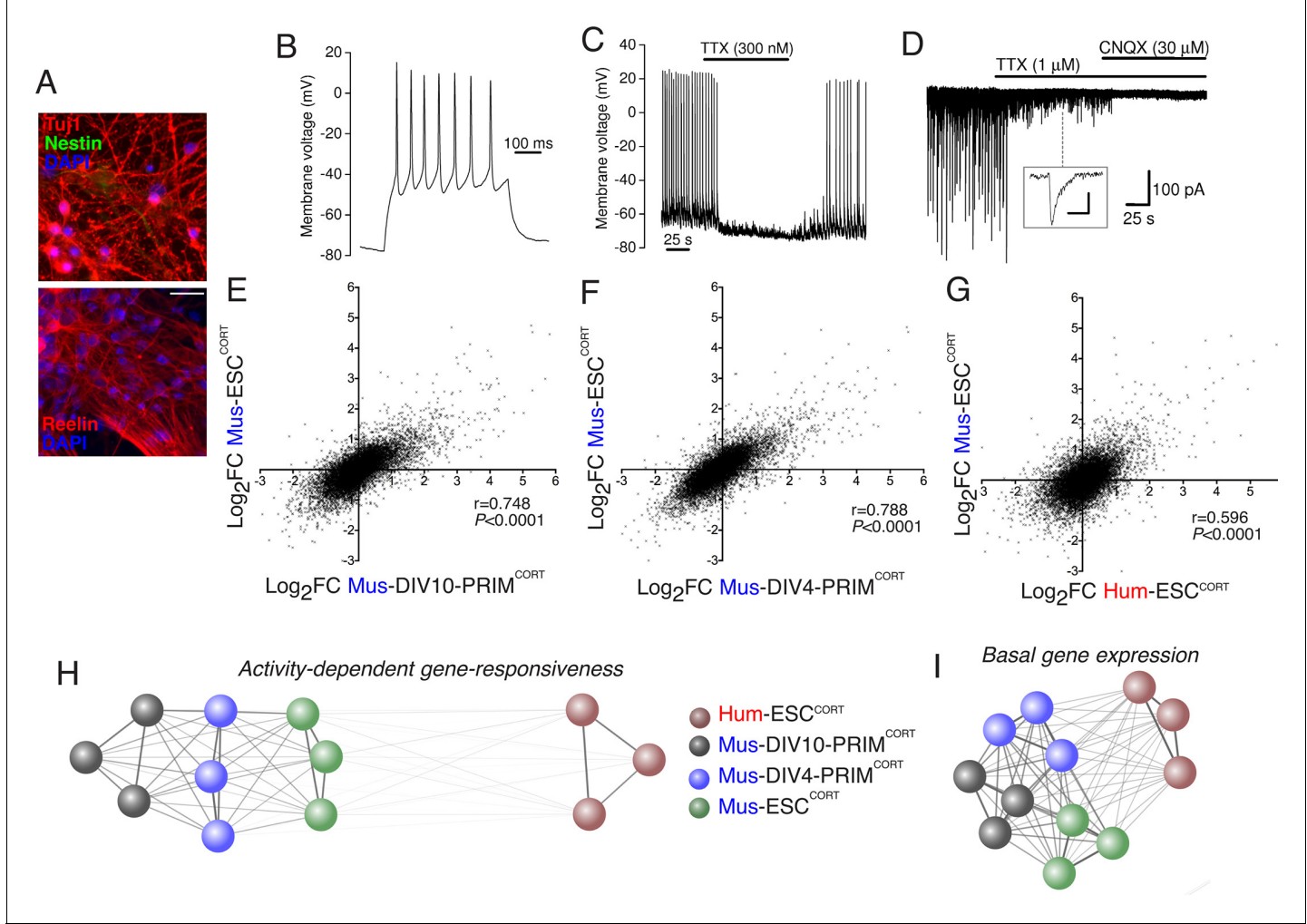

**Figure 2.** Stem cell origin does not substantially impact on activity-dependent gene responsiveness. (A) Example immunofluorescence pictures of Mus-ESC[CORT]-neurons stained for neuronal markers Tuj1 (upper) and Reelin (lower). Note also absence of Nestin staining (upper), a marker of undifferentiated neural precursor cells. Scale = 20 μm. (B) Example trace of a burst of action potentials (APs) induced in Mus-ESC[CORT]-neurons by current injection (see Materials and methods). (C) Example trace illustrating spontaneous TTX-sensitive AP firing. (D) Example trace illustrating spontaneous TTX-sensitive EPSCs, as well as TTX-insensitive, CNQX-sensitive miniature EPSCs (also see *inset*; scale bar: 20 pA, 5 ms). Activity returned upon wash out of TTX and CNQX (not shown). (E,F) Correlation of KCl/FPL-induced fold-change in the same 11,302 genes as in *Figure 1d,e* in Mus-ESC[CORT]-neurons vs. DIV10 (E) or DIV4 (F) Mus-PRIM[CORT]-neurons. (G) Correlation of KCl/FPL-induced fold-change in 11,302 ortholog pairs in Mus-ESC[CORT]-neurons vs. Hum-ESC[CORT]-neurons. (H) A connection map generated in Cytoscape (*Shannon et al., 2003*) illustrating the relative determination coefficients ($R^2$) between the fold-inductions of the 11,302 genes studied in each of the three biological replicates of the experiments performed in each of the four different neuronal preparations. The thickness of the connecting line and the attractive force of the connecting nodes are both directly proportional to $R^2$, against a background of constant inter-node repulsion. Note that all mouse neurons of differing developmental stage and origin (primary vs. ES cell) cluster strongly together, with the Hum-ESC[CORT]-neuronal replicates clustering away from them. (I) A connection map generated as for (H) but illustrating the relative determination coefficients ($R^2$) between the basal expression levels (FPKM) of the 11,302 genes studied in each of the three biological replicates of the experiments performed in each of the four different neuronal preparations.

The following source data and figure supplement are available for figure 2:

**Source data 1.** Data set relating to *Figure 2e–g*.

**Source data 2.** Data set relating to *Figure 2—figure supplement 1e*.

**Source data 3.** Data set relating to *Figure 2—figure supplement 1b–d*.

**Figure supplement 1.** Differential gene inducibility is not linked to basal levels of gene expression.

that the weaker correlation with the hESC$^{CORT}$-neurons may indeed be due to some species-specific differences in neuronal gene activity-responsiveness, rather than differences in developmental stage or cellular origin.

As a final genome-wide comparison we wanted to determine whether activity-dependent gene responsiveness showed a higher or lower degree of divergence than basal gene expression levels, and whether differences in basal gene expression were any predictor of differential activity-dependent gene responsiveness. Comparison of basal expression levels between Hum-ESC$^{CORT}$-neurons and mouse neurons (DIV10 Mus-PRIM$^{CORT}$-neurons, DIV4 Mus-PRIM$^{CORT}$-neurons, and Mus-ESC-$^{CORT}$-neurons) revealed correlation coefficients of 0.714, 0.711, and 0.710 (*Figure 2—figure supplement 1b–d*, *Figure 2—source data 3*). This correlation is substantially stronger than the that observed when comparing gene fold-change after KCl stimulation (0.480, 0.526, 0.595, *Figures 1d*, *2g*, *Figure 1—figure supplement 1g*). Moreover, we performed a similar clustering analysis as in *Figure 2h* which illustrates this graphically: i.e. while the basal expression profile of Hum-ESC$^{CORT}$-neurons does not cluster as closely as the three mouse neuronal populations do to each other, the degree of difference is less than that observed in *Figure 2h* (*Figure 2i*). Therefore basal neuronal gene expression shows less divergence than the responsiveness of genes to depolarization. We also investigated whether gene differential responsiveness to KCl (DRI) in human vs. mouse neurons has any relationship with the relative basal expression of that gene in human vs. mouse neurons. For each of the 11,302 orthologous pairs, we plotted the Log$_2$(DRI) Hum-ESC$^{CORT}$- vs. DIV10 Mus-PRIM-$^{CORT}$-neurons (i.e. DRIs from *Figure 1—figure supplement 1e*) against the Log$_2$(DBEI), where DBEI (differential basal expression index) is the ratio of basal expression in Hum-ESC$^{CORT}$- vs. DIV10 Mus-PRIM$^{CORT}$-neurons (*Figure 2—figure supplement 1e*, *Figure 2—source data 2*). As can be seen, there is no link between a gene's relative responsiveness to depolarization in human vs. mouse neurons, and its relative basal expression levels in human vs. mouse neurons. Moreover, if we consider the 657 genes where Log$_2$(DRI)>1, the standard deviation of their respective Log$_2$(DBEI), 1.45, is similar to the standard deviation of Log$_2$(DBEI) across all 11,302 genes (1.39), suggesting that there is no dramatic change in divergence of basal gene expression *regardless of direction*, in genes where DRI>1. Thus, evolutionary differences in activity-dependent gene responsiveness are not substantially attributable to differences in basal expression.

We hypothesized that species-specific gene-responsiveness to neuronal activity could be in part due to evolutionary divergence in genetic sequence, such as the promoters and distal enhancers that mediate activity-dependent changes in gene expression (*West and Greenberg, 2011*; *Kim et al., 2010*). To investigate whether genetic sequence could be sufficient to quantitatively influence species-specific gene inducibility, we utilized neurons cultured from the Tc1 transchromosomic mouse strain which carries a copy of human chromosome 21 (Hsa21), albeit with approximately 10% deleted (*O'Doherty et al., 2005*; *Gribble et al., 2013*). Thus, regulation of Hsa21 genes, plus their mouse 1:1 ortholog, could be studied side-by-side in the same cellular environment of a mouse primary neuron. We cultured cortical neurons from Tc1 embryos to DIV10, stimulated ± KCL/FPL, performed RNA-seq, and developed a workflow that distinguished between human and mouse RNA-seq reads, discarding any reads that were ambiguous. Once the initial criterion of a perfect sequence match of the RNA-seq read to either Hsa21 or the mouse genome was met, only a further 0.055% of reads were discarded due to 100% sequence conservation between species. Thus, the expression level of human and mouse genes carried in the Tc1 mouse can be analysed ± stimulation (*Figure 3a*). We first verified that the mouse genes in Tc1 neurons responded similarly to wild-type mouse neurons of the same age (*Figure 3—figure supplement 1a*, *Figure 3—source data 1*, r = 0.900).

72 orthologous pairs were then focussed on, where the Hsa21 ortholog met an expression cut-off of >100 reads. For each pair, we calculated the 'Tc1 DRI' (fold-change (human ortholog)/fold-change (mouse ortholog)) as a measure of whether responsiveness of the orthologs was different or similar, and plotted it against the gene's DRI calculated from fold-induction in *separate* human/mouse neuronal preparations: fold-change (Hum-ESC$^{CORT}$-neurons) / fold-change (DIV10 Mus-PRIM$^{CORT}$-neurons). Importantly, we observed a significant correlation between these DRIs (p<0.0001, r = 0.730, *Figure 3b*, *Figure 3—source data 2*), meaning that differences in gene regulation observed in separate human and mouse cortical neuron preparations ('separate DRI') are broadly recapitulated when the orthologs are studied side-by-side in the same cellular environment ('Tc1 DRI'). This points to the DNA sequence as a contributor to quantitative species-specific gene responsiveness. An

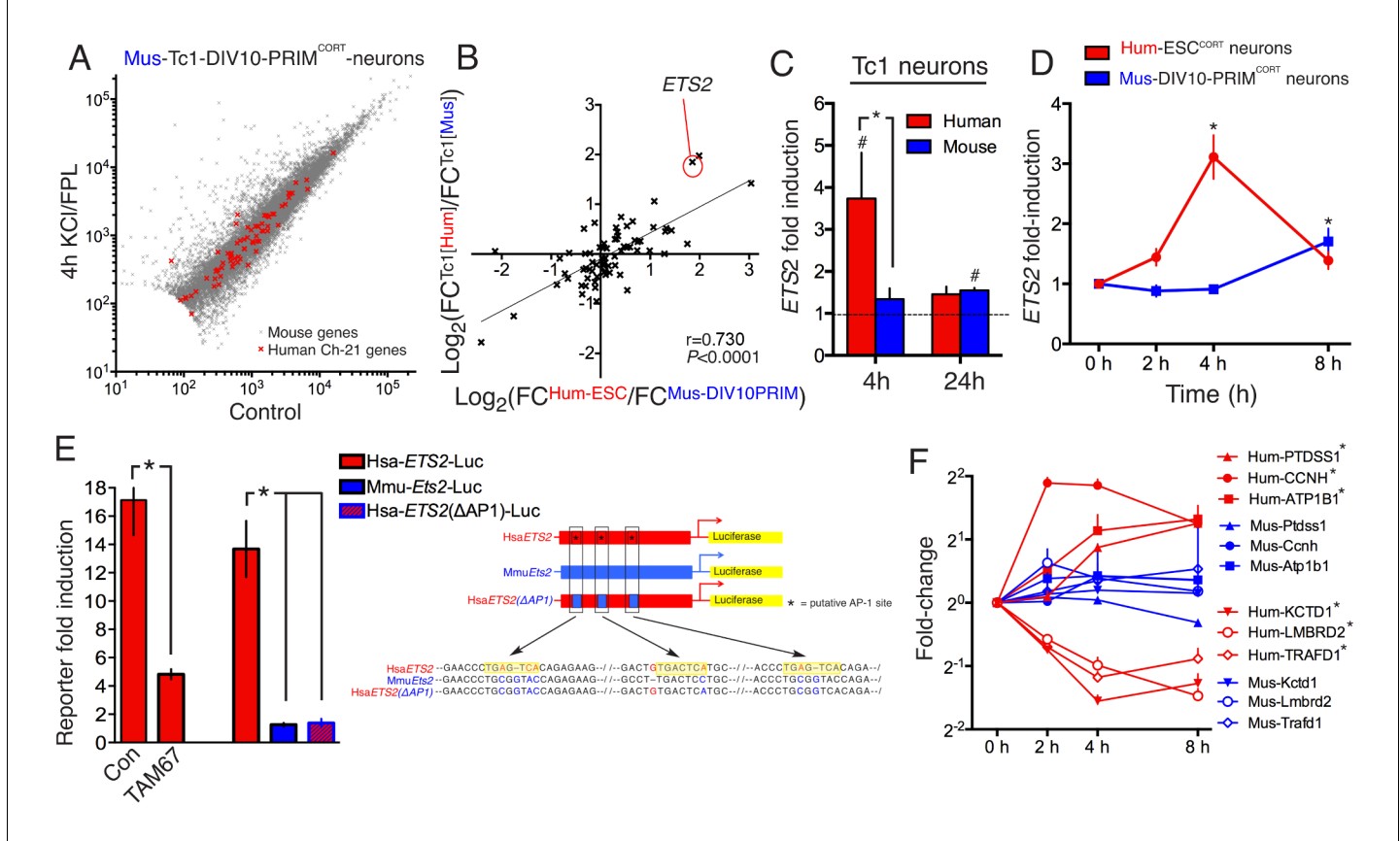

**Figure 3.** DNA sequence is a contributing factor to species-dependent gene responsiveness to neuronal activity. (**A**) Analysis of gene expression changes induced by KCl/FPL in mouse Tc1 neurons. Mouse genes are shown in grey, human chromosome-21 (Hsa-21) genes in red. (**B**) The graph concerns 72 orthologous pairs whose human ortholog is on Hsa-21 and carried by the Tc1 mouse strain, and which meets expression cut-off (100 reads). For each pair, the differential responsiveness index (DRI) was calculated from the species-separated Tc1 neuron RNA-seq data (see text), and plotted against the DRI calculated from separate neuronal preparations (Hum-ESC$^{CORT}$-neurons vs. DIV10 Mus-PRIM$^{CORT}$-neurons, as was done in *Figure 1—figure supplement 1e*). (**C**) KCl/FPL-induced fold induction of the human (red) and mouse (blue) orthologs of Hsa-21 gene *ETS2*, analysed side-by-side in Tc1 neurons. *p=0.035. unpaired t-test; # indicates p=0.0464, 0.0026 (left to right) vs. unstimulated control (4 hr, n = 7; 24 hr, n = 3). (**D**) Kinetics of KCl/FPL-induced *ETS2* induction in Hum-ESC$^{CORT}$-neurons vs. DIV10 Mus-PRIM$^{CORT}$-neurons. *p=0.0017, 0.004 (left to right), unpaired 1-way ANOVA vs. control (Hum-ESC$^{CORT}$-neurons: n = 3 (2 hr, 8 hr), n = 6 (4h); DIV10 Mus-PRIM$^{CORT}$-neurons: n = 4 (**E**) The indicated firefly luciferase reporters based on human (Hsa) or mouse (Mmu) *ETS2* promoters (See (Right) for schematic) plus a pTK-renilla control were transfected into mouse cortical neurons, treated ± KCl/FPL after which firefly:renilla luciferase ratio was measured, and fold-induction calculated. For comparing the effect of TAM67, a control vector (encoding β-globin) was used. *p=0.0015, 0.0015, 0.0015, 2-tailed paired *t*-test (n = 3). (**F**) Kinetics of gene regulation of 6 human:mouse ortholog pairs that show quantitative differences in activity-dependent regulation in Hum-ESC$^{CORT}$-neurons vs. all mouse neuronal preparations from prior RNA-seq analyses. Analysis was performed in Hum-ESC$^{CORT}$-neurons vs. DIV10 Mus-PRIM$^{CORT}$-neurons (n = 3,4 respectively). At the timepoints studied, three of these pairs exhibit quantitatively higher induction in human neurons, and three stronger repression in human neurons. *p=<0.0001, 0.0004, 0.0005, <0.0001, <0.0001, 0.0073 (top to bottom, 2-way ANOVA, p-value corresponds to the main species effect).

The following source data and figure supplement are available for figure 3:

**Source data 1.** Data set relating to *Figure 3—figure supplement 1a*.

**Source data 2.** Data set relating to *Figure 3b*.

**Source data 3.** Data set relating to *Figure 3c–f*.

**Figure supplement 1.** Inducibility of mouse genes in wild-type and Tc1 neurons strongly correlate.

example of such a gene is *ETS2*, the human ortholog of which is induced more strongly and rapidly than the mouse ortholog (*Figure 3c,d*, *Figure 3—source data 3*). Interestingly, a region of its proximal promoter contains three human-specific AP-1 sites (*Figure 3e*, right) and was found to confer activity inducibility on a luciferase reporter, unlike the corresponding region of the mouse *Ets2* promoter (*Figure 3e*, left). Moreover, expression of inhibitory AP-1 mutant TAM67 (*Brown et al., 1993*) inhibited Hum-*ETS2*-luciferase reporter induction, and mutation of the three AP-1 sites to the corresponding mouse sequence abolished inducibility (*Figure 3e*). Thus, the *ETS2* proximal promoter contains functional human-specific AP-1 sites, potentially contributing to the stronger and more rapid induction of the human ortholog in neurons

Although the focus of this study is understanding the extent and causes of differences in gene activity-responsiveness in human vs. mouse neurons, it should be noted that qualitative gene up/down regulation is well conserved. For example, of the top 100 most strongly induced human genes in Hum-ESC$^{CORT}$-neurons (FC>3.5-fold, Padj<0.05), 85 are also induced in Mus-ESC$^{CORT}$-neurons (FC>1.5-fold, Padj<0.05). Of the remaining 15 genes, 6 are induced in DIV4 and/or DIV10 Mus-PRIM$^{CORT}$-neurons, taking the total to 91/100. Conserved inducibility remains high even at a lower fold-change cut-off: e.g. 409 out of 475 genes induced >2-fold (Padj<0.05) in Hum-ESC$^{CORT}$-neurons are also induced (FC>1.5-fold, Padj<0.05) in one or more of the mouse neuronal preparations. Although we confirmed examples of genes regulated both up and down at 4h in Hum-ESC$^{CORT}$-neurons but not in mouse neurons, by performing a timecourse in Hum-ESC$^{CORT}$- and DIV10-Mus-PRIM-$^{CORT}$ neurons (*Figure 3f*), we hesitate to label these as definitively human-specific activity-response genes: there may be time-points, stimulation paradigms, or developmental stages that do expose activity-dependency on mouse neurons.

Nevertheless, our study points to quantitative differences in activity-dependent gene-responsiveness in developing human neurons, compared to their mouse counterpart, mediated at least in part by DNA sequence divergence. It is possible that these differences have a functional impact on the physiological response of human neurons to electrical activity. More generally, given that diverse signal dependent transcriptional programs mediate cellular responses to physiological stimuli, as well as the effects (and side-effects) of pharmaceutical agents (*Bell and Hardingham, 2011*; *Gupta et al., 2012*; *Bell et al., 2011*; *Hardingham and Lipton, 2011*; *Baxter et al., 2011*), this study points to the value of employing human neurons as a tool to help bridge the translational gap between rodent studies, and human neurophysiology.

# Materials and methods

## Generation of cortical-patterned neurons from human ES cells

A detailed description of the derivation of hESC$^{CORT}$-neurons, including immunohistochemical validation of cortical markers as well as electrophysiological characterization can be found in recent studies from the authors (*Bilican et al., 2014*; *Livesey et al., 2014*). Briefly, the hESC line H9 was used, and obtained from WiCell (Madison, WI) under full ethical / IRB approval of the University of Edinburgh. hESCs were maintained on CF-1 irradiated mouse embryonic fibroblasts, neurally converted to anterior-patterned neural precursor cells (NPCs) in suspension in chemically defined medium as described (*Stacpoole et al., 2011*), whereupon neural rosettes were mechanically isolated, dissociated and plated for proliferative expansion. For differentiation of anterior NPCs, they were plated in default media (A-DMEM/F12, 1% P/S, 0.5% Glutamax, 0.5% N2, 0.2% B27, 2 µg/mL Heparin (Sigma)) on poly-D-lysine (Sigma), laminin (Sigma), fibronectin (Sigma) and matrigel coated coverslips for differentiation and fed twice a week. Default media was supplemented with 10 µM forskolin (Tocris) in weeks 2 and 3. From week 4 onwards forskolin was removed and default media was supplemented with 5 ng/mL BDNF and 5 ng/mL GDNF. Experiments were performed 5 weeks after the differentiation of anterior NPCs was commenced.

## Generation of cortical-patterned neurons from mouse ES cells

Mouse ES cells (E14tg2α) were maintained in GMEM (Sigma) supplemented with 2-mercaptoethanol (Gibco), non-essential amino acids (Gibco), glutamine, pyruvate, 10% foetal calf serum (Gibco) and 100 units/ml LIF (made in-house) on gelatinised tissue culture flasks. Monolayer neural differentiation is described elsewhere (*Pollard et al., 2006*) and modified here to include cyclopamine. Briefly, ES

cells were washed to remove all traces of serum and then plated at a density of $1 \times 10^4$ cells/cm$^2$ onto gelatin-coated tissue culture plastic in N2B27 serum-free medium. N2B27 consists of a 1:1 ratio of DMEM/F12 and Neurobasal media (Gibco) supplemented with 0.5% modified N2 (made in house as described (*Pollard et al., 2006*), 1% B27 (Gibco), 2- mercaptoethanol (Gibco), 2 mM L-Glutamine, and 1uM Cyclopamine (Sigma). Medium was changed every day. Cells were cultured for 9 days under these conditions before transferring to terminal differentiation conditions. Terminal differentiation of mouse NPCs into dorsal forebrain (cortical) neurons was achieved as described elsewhere (*Gaspard et al., 2008*) by withdrawal of cyclopamine from day 9 NPCs dissociated into single cells, with Accutase (Life technologies), and re-plated on Matrigel (SLS, 354230; 1 in 100 dilution), Fibronectin 20 mg/ml (F2006 – Sigma-Aldrich), Laminin 10 mg/ml (L2020 – Sigma-Aldrich) coated coverslips at a density of 30,000 cells per 0.3 cm$^2$. Addition of 5 µM AraC on day 10, for 24 hr, was used to remove residual proliferating cells. Differentiating cortical neurons were maintained for a further 12–14 days, with media changes every 2–3 days, in Advanced DMEM/F12 (Invitrogen) containing: 1% N-2 supplement (Invitrogen), 1% B27 supplement (Invitrogen), 1% penicillin/streptomycin (Invitrogen), 0.5% GlutaMAX (Invitrogen), and 5 mg/ml heparin (Sigma), supplemented with 1 mM N-acetyl cysteine (Sigma) and 10 ng/ml BDNF (R&D Systems) and finally processed for RNA-seq and electrophysiological studies. For cell culture IHC, cells were fixed in 4% formaldehyde for 20 min at room temperature, washed with PBS and permeabilized with the detergent NP40 (Life Technologies). Cells were subsequently incubated in primary antibodies over night at 4°C. The next day, cells were washed with PBS and incubated with the appropriate secondary antibody at room temperature for 2 hr. Cells were then mounted using the mounting medium Vectashield (Vector Labs).

## Mouse primary cortical neuronal culture from wild-type and Tc1 mice

Cortical mouse neurons were cultured from either wild-type CD1 mice or Tc1 mice described (*Martel et al., 2012*) at a density of between $9–13 \times 10^4$ neurons per cm$^2$ from E17.5 mice with Neurobasal growth medium supplemented with B27 (Invitrogen, Paisley, UK). Stimulations of cultured neurons were done after a culturing period of 9–11 days during which neurons develop a network of processes, express functional NMDA-type and AMPA/kainate-type glutamate receptors, and form synaptic contacts. For Tc1 mouse genotyping, DNA was isolated from cerebellum using Qiagen QIAamp DNA Mini Kit following manufacturer's instructions. Genotyping reactions were performed using the following primers: D21S55 Forward =5'-GGT TTG AGG GAA CAC AAA GCT TAA CTC CCA-3', reverse =5'-ACA GAG CTA CAG CCT CTG ACA CTA TGA ACT-3'; Myo Forward =5'-TTA CGT CCA TCG TGG TGG ACA GCAT-3', reverse =5'- TGG GCT GGG TGT TAG TCT TAT-3'. D21S55 recognizes the Tc1 allele (product =208 bp) whereas Myo recognizes both Tc1 and WT alleles (product =245 bp).

## Stimulation of mouse cortical and hESC$^{CORT}$-neurons

Prior to stimulation, neurons were first placed overnight into a minimal defined medium (*Papadia et al., 2005*) containing 10% MEM (Invitrogen), 90% Salt-Glucose-Glycine (SGG) medium ([*Bading et al., 1993*]; SGG: 114 mM NaCl, 0.219% NaHCO$_3$, 5.292 mM KCl, 1 mM MgCl$_2$, 2 mM CaCl$_2$, 10 mM HEPES, 1 mM Glycine, 30 mM Glucose, 0.5 mM sodium pyruvate, 0.1% Phenol Red; osmolarity 325 mosm/l,[*Papadia et al., 2005*]). KCl/FPL stimulations were performed as described previously (*Hardingham et al., 1999, 1997*). KCl/FPL stimulation involved neurons being exposed to 50 mM KCl by adding 0.41 volumes of KCl depolarization solution (*Bading et al., 1993*) (10 mM HEPES, pH 7.2, 170 mM KCl, 1 mM MgCl$_2$, 2 mM CaCl$_2$) in the presence of 5 µM FPL64176 (Tocris), plus an NMDA receptor antagonist (MK-801, 5 µM) to prevent any excitotoxicity. As described recently (*Bilican et al., 2014*), hESC$^{CORT}$-neurons respond strongly and uniformly to KCl/FPL treatment, with over 80% of hESC$^{CORT}$-neurons exhibiting a >10-fold increase in [Ca$^{2+}$], and the remainder showing at least a 5-fold increase.

## Calcium imaging

Ca$^{2+}$ imaging was performed as described (*Hardingham et al., 1997*; *Soriano et al., 2008*) at 37°C in aCSF (150 mM NaCl, 3 mM KCl, 10 mM HEPES, 2 mM CaCl2, 1 mM MgCl$_2$, 1 mM glucose). Briefly, cells were loaded with 11 µM Fluo-3 AM (from a stock solution of 2.2 mM Fluo-3 dissolved in

anhydrous DMSO containing 20% (w/v) Pluronic detergent) for 30 min at 37°C. Fluo-3 fluorescence images (excitation 472 ± 15 nm, emission 520 ± 15 nm) were taken at one frame per 5 s using a Leica AF6000 LX imaging system, with a DFC350 FX digital camera. To calibrate images, Fluo-3 was saturated by adding 50 µM ionomycin to the perfusion chamber (to obtain $F_{max}$) and quenched with 10 mM $MnCl_2$ + 50 µM ionomycin to levels corresponding to 100 nM $Ca^{2+}$ (*Minta et al., 1989*), which was in turn used to calculate $F_{min}$. Free $Ca^{2+}$ concentrations were calculated from fluorescence signal (F) according to the equation $[Ca^{2+}] = Kd(F - F_{min})/(F_{max} - F)$, and expressed as a multiple of the Kd of Fluo-3 (which is approximately 315 nM).

## Electrophysiology

The whole-cell patch configuration was used to record membrane currents and voltage deflections as previously described (*Bilican et al., 2014*; *James et al., 2014*). Membrane potential data are corrected for liquid junction potential (+14 mV). Current and voltage measurements were typically low-pass filtered online at 2 kHz, digitized at 10 kHz and recorded to computer using the WinEDR V2 7.6 Electrophysiology Data Recorder (J. Dempster, Department of Physiology and Pharmacology, University of Strathclyde, UK).

## RNA sequencing

Neurons were treated with KCl/FPL as described above. Cells were lysed and RNA extracted at 4h post-stimulation. For both mouse and human-based experiments, three independent biological replicates were performed, and analysed in parallel. Three replicates were deemed sufficient based on previous transcriptome studies on activity-dependent gene expression in which we successfully identified differentially regulated genes with this number of replicates (*Papadia et al., 2008*). Total RNA was assessed for quality (Agilent Bionalyzer) and quantity (Invitrogen Qubit) before library preparation. Illumina libraries were prepared from 1 µg of total RNA using TruSeq RNA Sample Prep Kit v2 with a 10 cycle enrichment step as per the manufacturer's recommendations. Final libraries were pooled in equimolar proportions before Illumina sequencing on a HiSeq 2500 platform using 75 base paired-end reads. Raw reads were processed using RTA 1.17.21.3 and Casava 1.8.2 (Illumina). Reads were mapped to the mouse (mm10) and human (hg38) reference genomes using version 2.4.0i of the STAR RNA-seq aligner (*Dobin et al., 2013*). A table of per-gene read counts was generated from the mapped reads with featureCounts version 1.4.6-p2 (*Liao et al., 2014*), using gene annotations from Ensembl version 82. Differential expression analysis was then performed using DESeq2 (R package version 1.10.0) (*Love et al., 2014*). Raw data will be deposited in the European Nucleotide Archive in advance of publication.

### Species-specific sorting of Tc1 neuron RNA-seq data

Given a set of RNA-seq reads that have derived from transcripts of both the mouse genome and human chromosome-21, we implemented a sorting procedure to assign reads to their true species of origin. In the balance between precision and recall, our strategy is conservative, in that it aims foremost to minimise the number of reads allocated to the incorrect species; but we simultaneously seek to maximise the number of reads that can be unambiguously assigned to the correct species. The sorting criteria detailed below were chosen to effect this goal. In this species-specific sorting (SSS) procedure, reads are first mapped to the genomes of each species with the STAR RNA-seq aligner. At this stage multi-mapping alignments are allowed (–outFilterMultimapNmax 10000), but only those with an alignment score equal to the maximum (–outFilterMultimapScoreRange 0). Subsequently, for each RNA-seq read the alignments of that read to each genome are compared. If the read has alignments to the mouse genome, but no alignments to the human chromosome-21 exist, the read is provisionally assigned to the mouse; note, however, the further requirements given below for a read to be finally allocated to this genome. Similarly, if alignments to human chromosome-21 exist, but there are no alignments to the mouse genome, the read is provisionally assigned to the human. If alignments to both genomes exist, these alignments are examined in more detail. Firstly, any read which aligns multiple times to either genome is discarded, since in this initial conservative strategy their exclusion removes a potential source of ambiguity. Next, the number of mismatched bases between the mapped read and each genome is examined. If the number of mismatches is smaller for one species, the read is provisionally allocated to that genome. If the number of

mismatches for each species is equal, a further check is made on the structure of the alignments; a successful alignment is required to span the full length of the read (without any clipping of bases), and, if the alignment spans an intron, at least 5 bases are required to align to the exons on either side of the boundary. If these criteria are satisfied for the alignment to one genome, but not to the other, then the read is provisionally assigned to the first genome. If the criteria are satisfied for the alignments to both genomes, the read is rejected as ambiguous; it cannot be assigned with confidence to one species rather than the other. The read is also rejected if the structural criteria are not satisfied for the alignments to either genome.

At this stage, a read has either been provisionally allocated to one species, or has been rejected. In the former case, a final set of checks are made on the provisional alignment. In this conservative strategy, these are that there should be no mismatches between the mapped read and the genome, and that the structural criteria outlined above are satisfied (if this has not already been confirmed). The outcome of this procedure is that all reads successfully assigned to one species or the other have a single, full-length alignment to that species' genome, with no mismatched bases. Subsequently, for each sample, per-gene read counts were summarised using featureCounts. Relative expression levels of genes are expressed as fragments per million reads per kilobase of message (FPKM). Within the SSS-workflow, only reads that are unambiguously attributed to a particular species are used as the denominator in the FPKM calculation. The value for the length of message for a particular gene refers to the maximum transcript length. Where gene length data is given, this refers to the number of nucleotides contained within the union of all exons of all transcripts of the gene, including 5' and 3' UTRs. Differential expression analysis on data sets was performed using DESeq2.

## Plasmids and reporter assays

A region of the human *ETS2* promoter (NM_001256295 -1080 to -19, or NM_005239.5, positions -1622 to -542) was generated by PCR amplification using Stratagene UltraII Fusion HS DNA Polymerase kit and subcloned into pGL4.10. All other promoter sequences (both wild-type and mutant) were obtained as synthetic clones from Life Technologies (Geneart) and then subcloned into pGL4.10. This included the corresponding mouse *Ets2* promoter sequence (ENSMUST00000023612 -1055 to -65, or NM_011809.3 -1387 to -397) and a mutated form of the human *ETS2* promoter with the putative AP-1 sites mutated to the corresponding mouse sequence (*ETS2*(ΔAP1)) was synthesized. The following plasmids have been described previously: pcDNA3-cJun(3–122) (TAM-67) (*Brown et al., 1993*). Firefly luciferase-based reporter gene constructs were transfected (Lipofectamine 2000) into DIV8 mouse cortical neurons as along with a Renilla expression vector (pTK-RL) control. The number of biological replicates were performed based on previous studies where site-directed mutagenesis was performed on reporter constructs to assess activity-dependent promoter regulation (*Papadia et al., 2008*). Neurons were stimulated with KCl/FPL (where appropriate) 24 hr after transfection. Luciferase assays were performed using the Dual Glo assay kit (Promega) with Firefly luciferase-based reporter gene activity normalized to the Renilla control (pTK-RL plasmid) in all cases.

## RNA extraction and qPCR

For RNA-seq and qPCR, RNA was isolated using the Roche HP RNA Isolation kit according to manufacturer's instructions. For qPCR, cDNA was synthesized from 1– 3 μg RNA using Roche Transcriptor cDNA Synth kit. Briefly, RNA was diluted in RNase-free water and mixed on ice with 1 μl oligo-dT primers (50 pmol/μl), 2 μl random hexamers (600 pmol/μl), 4 μl Transcriptor RT Reaction Buffer (5x), 0.5 μl Protector RNase Inhibitor (40 U/μl), 2 μl Deoxynucleotide Mix (10 mM each), 0.5 μl Transcriptor Reverse Transcriptase and made up to 20 μl with RNAase-free water. Reaction mixtures were incubated for 2 min at 25°C, 40 min at 42°C and 5 min at 95°C. Resultant cDNA was stored at −20°C.

qPCR was performed in an Mx3000P QPCR System (Stratagene) using Roche FS Universal SYBR Green MasterRox. Briefly, cDNA (equivalent to 6 ng of starting RNA) was mixed with concentrated master mix (2x) and appropriate forward and reverse primers (20 nM) and made up to a final volume of 15 μl with RNase-free water. Technical replicates as well as no template and no RT controls were included in each run. Gene expression levels were normalized in all cases to *Gapdh* housekeeping control. Primer sequences used are as follows:

| Primer | Sequence |
|---|---|
| Human-*ARC* | F: 5'-CTGAGCCACCTAGAGGAGTACT-3'. R: 5'-AACTCCACCCAGTTCTTCACGG-3' |
| Mouse-*Arc* | F: 5'-GCTGGAAGAAGTCCATCAAGGC-3'. R: 5'-ACCTCTCCAGACGGTAGAAGAC-3' |
| Human-*BDNF* | F: 5'-AGCTGAGCGTGTGTGACAGT-3'. R: 5'-ATGGGATTGCACTTGGTCTC-3' |
| Mouse-*Bdnf* | F: 5'-AAAGTCCCGGTATCCAAAGG-3'. R: 5'-CTTATGAATCGCCAGCCAAT-3' |
| Human-*CCNH* | F: 5'-CGATGTCATTCTGCTGAGCTTGC-3'. R: 5'-TCTACCAGGTCGTCATCAGTCC-3' |
| Mouse-*Ccnh* | F: 5'-ACTTGCCTGTCACAGTTACTGGA-3'. R: 5'-GAATGACACCGCTCCAGCTTCT-3' |
| Human-*ETS2* | F: 5'-CAACTCCTTTTCAGAGATCAGG-3'. R: 5'-TTTCATCAAGACCCCTACCG-3' |
| Mouse-*Ets2* | F: 5'-TCACGTAAAGGGAGATGTGTCG-3'. R: 5'-TGCTCTGTCTGTGCTTCTGG-3' |
| Human-*FOS* | F: 5'-CTACCACTCACCCGCAGACT-3'. R: 5'-AGGTCCGTGCAGAAGTCCT-3' |
| Mouse-*Fos* | F: 5'-CCATGATGTTCTCGGGTTTC-3'. R: 5'-TGGCACTAGAGACGGACAGA-3' |
| Human-*FOSB* | F: 5'-GCCGGGAACGAAATAAACTA-3'. R: 5'-CACCAGCACAAACTCCAGAC-3' |
| Mouse-*Fosb* | F: 5'-AGGGAGCTGACAGATCGACTT-3'. R: 5'-CTTCGTAGGGGATCTTGCAG-3' |
| Human-*FOSL2* | F: 5'-ACACCCTGTTTCCTCTCCG-3'. R: 5'-GATGGTGGGGATGAATGCAC-3' |
| Mouse-*Fosl2* | F: 5'-CGGGAACTTTGACACCTCGT-3'. R: 5'-AGGGATGTGAGCGTGGATAG-3' |
| Human-*GAPDH* | F: 5'-CTTCACCACCATGGAGAAGGC-3'. R: 5'-GGCATGGACTGTGGTCATGAG-3' |
| Mouse-*Gapdh* | F: 5'-GGGTGTGAACCACGAGAAAT-3'. R: 5'-CCTTCCACAATGCCAAAGTT-3' |
| Human-*NPAS4* | F: 5'-CCTGCATCTACACTCGCAAG-3'. R: 5'-CTCGCTCACACTCTCAGACA-3' |
| Mouse-*Npas4* | F: 5'-AGGGTTTGCTGATGAGTTGC-3'. R: 5'-CCCCTCCACTTCCATCTTC-3' |
| Human-*PER1* | F: 5'-TCAACTGCCTGGACAGCATCCT-3'. R: 5'-TCAGAGGCTGAGGAGGTGGTAT-3' |
| Mouse-*Per1* | F: 5'-GAAACCTCTGGCTGTTCCTACC-3'. R: 5'-AGGCTGAAGAGGCAGTGTAGGA-3' |
| Human-*SIK1* | F: 5'-CTCAGAGAGGGCAGAGGTGA-3'. R: 5'-ATGCATAAACGTCAGCAGCA-3' |
| Mouse-*Sik1* | F: 5'-GTGCCATCCAAACACCTCTG-3'. R: 5'-TGTCTGGAGAGTAAGCGGTA-3' |
| Human-*SLC2A3* | F: 5'-TGCCTTTGGCACTCTCAACCAG-3'. R: 5'-GCCATAGCTCTTCAGACCCAAG-3' |
| Mouse-*Slc2a3* | F: 5'-CCGCTTCTCATCTCCATTGTCC-3'. R: 5'-CCTGCTCCAATCGTGGCATAGA-3' |
| Human-*PTDSS1* | F: 5'- ATGTGATCACCTGGGAGAGG-3'. R: 5'- CCATTGCACAACAGGATGTC-3' |
| Mouse-*Ptdss1* | F: 5'- ACACAGTGCAAGCGTGTAGG-3'. R: 5'-AACCATGCCGTACAGACACA-3' |
| Human-*ATP1B1* | F: 5'-CCGCCAGGATTAACACAGAT-3'. R: 5'-TCCTCGTTCTTTCGGTTCAC-3' |
| Mouse-*Atp1b1* | F: 5'-CAGATTCCCCAGATCCAGAA-3'. R: 5'-CTGCACACCTTCCTCTCTCC-3' |
| Human-*KCTD1* | F: 5'-AGTCGGCCCAATATGTCAAG-3'. R: 5'-CCGATTCTGGATTCAGGGTA-3' |
| Mouse-*Kctd1* | F: 5'-AGTCGGCCCAATATGTCAAG -3'. R: 5'-TTCTGGATTCGGGGTATTTG -3' |
| Human-*LMBRD2* | F: 5'-TTGGGATAGCTGCTGCAAAT-3'. R: 5'-CTCCATGGCATCTTCCAAAT-3' |
| Mouse-*Lmbrd2* | F: 5'-GTGCGGCTTTAGGACTTGAG-3'. R: 5'-CAGCGGCAGTATGAAGACAA-3' |
| Human-*TRAFD1* | F: 5'-AAGGAGGGGAAGATTTTGGA-3'. R: 5'-GCCTGAGCACCTTACCAGAG-3' |
| Mouse-*Trafd1* | F: 5'-AGTCTGTGCCTGAGGCTGAT-3'. R: 5'-GAGAAGGGTTGCAGCTTGTC-3' |

## Statistical analysis, equipment and settings

Statistical testing of the RNA-seq data is described in that section. Other testing involved a 2-tailed paired Student's t-test., or a one- or two-way ANOVA followed by Sidak's post-hoc test, as indicated in the legends. Correlation coefficients were calculated assuming a Gaussian distribution of the data. Throughout the manuscript, independent biological replicates are defined as independently performed experiments on material derived from different animals.

## Acknowledgements

This work was supported by the MRC, the Wellcome Trust, the BBSRC and the Royal Society. SL is a Wellcome Trust Senior Research Fellow, GEH is a MRC Senior Non-Clinical Research Fellow. VLJT was supported by core funding from the Francis Crick Institute which receives its core funding from the UK Medical Research Council, Cancer Research UK and the Wellcome Trust. We thank Eleanor Coffey for supplying us with the TAM-67 vector.

## Additional information

### Funding

| Funder | Author |
| --- | --- |
| Biotechnology and Biological Sciences Research Council | Siddharthan Chandran Giles E Hardingham |
| Royal Society | Giles E Hardingham Siddharthan Chandran |
| Medical Research Council | Giles E Hardingham |
| Wellcome Trust | Giles E Hardingham |

The funders had no role in study design, data collection and interpretation, or the decision to submit the work for publication.

### Author contributions

JQ, JM, BB, KP, BTS, ML, GH, KB, RP, RR, OS, SL, Acquisition of data, Analysis and interpretation of data; OD, Conception and design, Acquisition of data, Analysis and interpretation of data; DM, Conception and design, Acquisition of data; SH, PCK, TIS, DJAW, SC, Conception and design, Analysis and interpretation of data; VLJT, Analysis and interpretation of data, Contributed unpublished essential data or reagents; EMCF, Acquisition of data, Analysis and interpretation of data, Contributed unpublished essential data or reagents; GEH, Conception and design, Analysis and interpretation of data, Drafting or revising the article

### Author ORCIDs

Owen Dando, http://orcid.org/0000-0002-6269-6408
Karolina Punovuori, http://orcid.org/0000-0003-0297-1225
T Ian Simpson, http://orcid.org/0000-0003-0495-7187
Victor LJ Tybulewicz, http://orcid.org/0000-0003-2439-0798
David JA Wyllie, http://orcid.org/0000-0002-4957-6049
Sally Lowell, http://orcid.org/0000-0002-4018-9480
Giles E Hardingham, http://orcid.org/0000-0002-7629-5314

### Ethics

Animal experimentation: Animals used in this study were treated in accordance with UK Animal Scientific Procedures Act (1986) and the work subject to local ethical review approval by the University of Edinburgh Ethical Review Committee. The relevant project licence is 7009008, and the use of genetically modified organisms approved by local committee reference SBMS 13_007.

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
