## [Decision Letter]

Thank you for submitting your article "Evidence for evolutionary divergence of activity-dependent gene expression in developing neurons" for consideration by *eLife*. Your article has been favorably evaluated by a Senior Editor and three reviewers, including Lynn Raymond (Reviewer #2) and a member of our Board of Reviewing Editors.

The reviewers have discussed the reviews with one another and the Reviewing Editor has drafted this decision to help you prepare a revised submission.

Summary:

In this manuscript the authors study the differential activity-dependent regulation of genes in mouse and human cells. Initially they derive neurons from human ES cells and use RNAseq to compare the programs of gene expression induced to those of mouse cortical neurons. These data are comprehensive and suggest significant differences in gene induction that could be important in the human. However, to address the concern that differences in the nature of the experimental material could also impact gene induction, the authors then perform a very interesting series of experiments in a mouse strain that carries a copy of human chromosome 21, allowing for comparison of activity-regulated Chr 21 genes in the same cells across species. Here the authors report one clear example (ETS2) that is far more activity-inducible from the human chromosome compared with the mouse. They map the difference in inducibility to differences in regulatory element sequence that create an AP-1 binding site only in the human. Overall this is a novel proof-of-principle example of species-specific differences in activity-dependent gene regulation in neurons.

Essential revisions:

1) From the data shown it remains unclear how gene expression from human and mouse compare with each other under basal conditions, without KCl treatment. This information is important for understanding whether the conservation and divergence between species is observed only after KCl treatment or in both treated *and* untreated conditions. One way the authors could answer this question is to perform cluster analyses and PCA or similar analyses as in Figure 2 on untreated samples; they could also focus on the genes that show divergence (DRI different from 1 cf Figure 1—figure supplement 1) and show whether their baseline levels are also divergent or conserved.

2) A great potential value of this manuscript is that it could serve as a database for future studies. To facilitate this, the authors should display in each data table the identity of the genes that show divergence between conditions (those that have DRI different from 1). For instance, there should be more explicit naming of the genes in the data set excel file linked to Figure 1—figure supplement 1, which is arguably constitute one of the most interesting lists. In addition, the columns could be more clearly labeled (for instance what is LN2 vs. DRI?).

---

## [Author Response]

[…] Essential revisions:

*1) From the data shown it remains unclear how gene expression from human and mouse compare with each other under basal conditions, without KCl treatment. This information is important for understanding whether the conservation and divergence between species is observed only after KCl treatment or in both treated and untreated conditions. One way the authors could answer this question is to perform cluster analyses and PCA or similar analyses as in Figure 2 on untreated samples; they could also focus on the genes that show divergence (DRI different from 1 cf Figure 1—figure supplement 1) and show whether their baseline levels are also divergent or conserved.*

We have performed several analyses to address the extent of divergence in basal gene expression across different neuronal preparations and species, as well as whether there is any relationship between differential responsiveness to KCl stimulation, and differential basal gene expression.

Comparison of basal expression levels between Hum-ESC^CORT^-neurons and mouse neurons (DIV10 Mus-PRIM^CORT^-neurons, DIV4 Mus-PRIM^CORT^-neurons, and Mus-ESC^CORT^-neurons) revealed correlation coefficients of 0.714, 0.711, and 0.710 (Figure 2—figure supplement 1). This correlation is substantially stronger than the that observed when comparing gene fold-change after KCl stimulation (0.480, 0.526,.595, Figure 1, Figure 2, Figure 1—figure supplement 1). Moreover, we performed a similar clustering analysis as in Figure 2 which illustrates this graphically: i.e. the basal expression profile of Hum-ESC^CORT^-neurons clusters far more closely to the three mouse neuronal populations (new Figure 2) compared to the activity-dependent gene responsiveness (Figure 2). Therefore, basal neuronal gene expression shows less divergence than the responsiveness of genes to depolarisation.

We also investigated whether gene differential responsiveness to KCl (DRI) in human vs. mouse neurons has any relationship with the relative basal expression of that gene in human vs. mouse neurons. For each of the 11,302 orthologous pairs, we plotted the Log_2_(DRI) Hum-ESC^CORT^-vs. DIV10 Mus-PRIM^CORT^-neurons (i.e. DRIs from Figure 1—figure supplement 1) against the Log_2_(DBEI), where DBEI (differential basal expression index) is defined as the ratio of basal expression in Hum-ESC^CORT^-vs. DIV10 Mus-PRIM^CORT^-neurons (Figure 2—figure supplement 1). As can be seen, there is no link between a gene's relative responsiveness to depolarisation in human vs. mouse neurons, and its relative basal expression levels in human vs. mouse neurons. Moreover, if we consider the 657 genes where Log_2_(DRI)>1, the standard deviation of their respective Log_2_(DBEI), 1.45, is similar to the standard deviation of Log_2_(DBEI) across all 11,302 genes (1.39), suggesting that there is no dramatic change in divergence of basal gene expression *regardless of direction*, in genes where DRI>1. Thus, evolutionary differences in activity-dependent gene responsiveness are not substantially attributable to differences in basal expression.

All source data for these calculations is included in the revised source data files.

*2) A great potential value of this manuscript is that it could serve as a database for future studies. To facilitate this, the authors should display in each data table the identity of the genes that show divergence between conditions (those that have DRI different from 1). For instance, there should be more explicit naming of the genes in the data set excel file linked to Figure 1—figure supplement 1, which is arguably constitute one of the most interesting lists. In addition, the columns could be more clearly labeled (for instance what is LN2 vs. DRI?).*

We have ensured that the identity of the gene is included in *all* the source data tables. We have also reviewed the column labels and edited them to make their meaning is clearer to the reader.